# Leveraging Visual Embeddings from Instagram for Credit Scoring of Informal Microbusinesses

## Abstract

Access to formal credit remains limited for informal microbusinesses in Latin America, forcing entrepreneurs to adopt predatory lending practices characterized by exorbitant interest rates. Several fintech startups have sought to address this challenge by developing alternative credit scoring methodologies leveraging artificial intelligence and non-traditional data sources. This paper presents a novel approach that applies computer vision techniques to Instagram images and videos from microbusiness accounts, extracting visual features to improve predictive models of creditworthiness. The proposed method utilizes pre-trained vision language models, such as CLIP and X-CLIP, to obtain visual embeddings. Subsequently, dimensionality reduction (UMAP) and clustering (KMeans) techniques are applied to derive discriminative features. In addition, two distinct architectures are introduced: a Fully Connected Neural Network (FCNN) processing CLIP embeddings, and a separate Convolutional Neural Network (CNN) directly analyzing image data, each generating predictive visual scores. Preliminary results show that visual embeddings improved AUC by 2.16 points and F1-score by 9.86 points, with visual features contributing 25.52% of predictive power—underscoring the potential impact of computer vision-based methodologies on financial inclusion among underserved communities.

## 1 Introduction

Micro, small, and medium enterprises (MSMEs) represent over 99% of businesses in Latin America, providing 60% of formal employment and contributing a quarter of the region's GDP Herrera (2020). Yet, most micro-entrepreneurs remain excluded from formal financial systems, facing a credit gap estimated at US$1.8 trillion across the region Altman & Sabato (2023).

In the absence of formal credit, many resort to informal "gota a gota" lenders who charge 20–40% monthly interest, perpetuating debt cycles and financial vulnerability Bloomberg Cities Nework (2020); Ramírez Hernández (2023); Bidart (2024). These conditions highlight the urgent need for inclusive, fair, and data-driven credit scoring solutions.

Alternative credit scoring methods leverage artificial intelligence (AI) and non-traditional data sources—especially the digital footprint of microbusinesses on social platforms. Instagram, in particular, is widely adopted by entrepreneurs in Latin America for branding and engagement Martínez-Estrella et al. (2023), offering a rich source of visual and behavioral data.

Recent advances in vision-language models like CLIP Radford et al. (2021) and X-CLIP Ni et al. (2022) enable the extraction of semantic embeddings from images and videos, capturing signals related to business operations, product quality, and customer interaction. These embeddings can enhance predictive models for credit risk.

This work introduces a novel approach to credit scoring that integrates computer vision pipelines applied to Instagram data. It combines pretrained embeddings, dimensionality reduction, unsupervised clustering, and supervised neural architectures (FCNN and CNN) to generate visual features. The aim is to improve model performance and support financial inclusion across underserved entrepreneurial communities in Latin America.

## 2 PROPOSED METHOD

The proposed methodology aims to improve alternative credit scoring models for informal microbusinesses by incorporating visual features derived from Instagram content. We introduce a series of computational techniques to process Instagram images and videos using pretrained vision-language models. To further enhance predictive accuracy, a supervised neural network is employed to refine the extracted visual embeddings into credit scores, which are further analyzed by an XGBoost. Each component of the methodology is described in detail in the following subsections.

### 2.1 DATASET AND EXPERIMENTAL SETUP

The dataset used in this study comprises Instagram account data from a total of 570 microbusiness who received microcredit financing from a fintech company operating in Colombia between 2022-2024.Borrowers are labelled *good payer* ($target = 1$) if they never exceeded the contractual grace period and *poor payer* ($target = 0$) otherwise, yielding **355 good** versus **215 poor** payers. The four largest business categories are *Services* (19%), *Beauty&Hygiene* (19%), *Handicrafts&Art* (19%) and *Restaurants&Food* (12%). Agriculture and Electronics together account for $<5\%$, which may introduce a sector-selection bias. To mirror real-world deployment, the dataset was split using an out-of-time strategy to ensure temporal validity: the training set includes 466 users whose loan disbursement dates precede those in the validation set (60 users), and subsequently, the test set (44 users).

Multiple data modalities were extracted from each user's Instagram account, including:

- **User Metadata:** Biography text, number of followers, number of accounts followed, business account status, total number of posts, business category, and availability of external contact links.
- **Post Information:** Number of highlights, post type (reel, carousel, or single image), geolocation tags, date of posting, number of likes, comments, views, tagged collaborators, and textual descriptions.
- **Visual Data:** Images and videos from user posts, analyzed through computer vision methodologies described in this paper.

This study evaluates whether visual features from Instagram images and videos can improve credit scoring with XGBoost. Starting from a baseline using structured user data, we test the impact of adding visual embeddings—extracted via clustering and a neural network—on predictive accuracy.

The hyperparameter tuning of the XGBoost model was performed using the Optuna optimization framework. To optimize model performance beyond standard metrics, we implemented a custom evaluation metric that balances multiple critical objectives for credit risk assessment. The evaluation metric is defined in Equation 1:

$$\mathcal{L} = -\beta_w \cdot \text{AUC} - \gamma \cdot \text{Accuracy} + \alpha \cdot \text{FPR} + \delta \cdot \text{Overlap} \tag{1}$$

where $\beta_w$, $\gamma$, $\alpha$, and $\delta$ are weights for each component. In particular, FPR defined as $FP/(TP + FP)$ quantifies the exposure to credit risk.

The overlap component measures the intersection between probability distributions of classes, calculated by fitting Gaussian kernel density estimators. to each class's predicted probabilities and computing the area where these distributions intersect: $\int_0^1 \min(f_0(x), f_1(x))dx$.

The function allows us to prioritize AUC maximization while penalizing high default rates. Additionally, it incorporates the overlap penalty that encourages better separation between class distributions, thereby improving the model's ability to clearly differentiate between good and poor payers. A Conversion Rate Constraint defined as the proportion of approved applicants $(TP + FP)/(TP + TN + FP + FN)$, ensures that the model approves a practical proportion of applicants. Because the data set is moderately imbalanced, the evaluation of model performance primarily focuses on **F1-score** to reflect the practical cost of misclassifying either class, and the **AUC-ROC** to provide a threshold-independent view of discriminative ability, two complementary

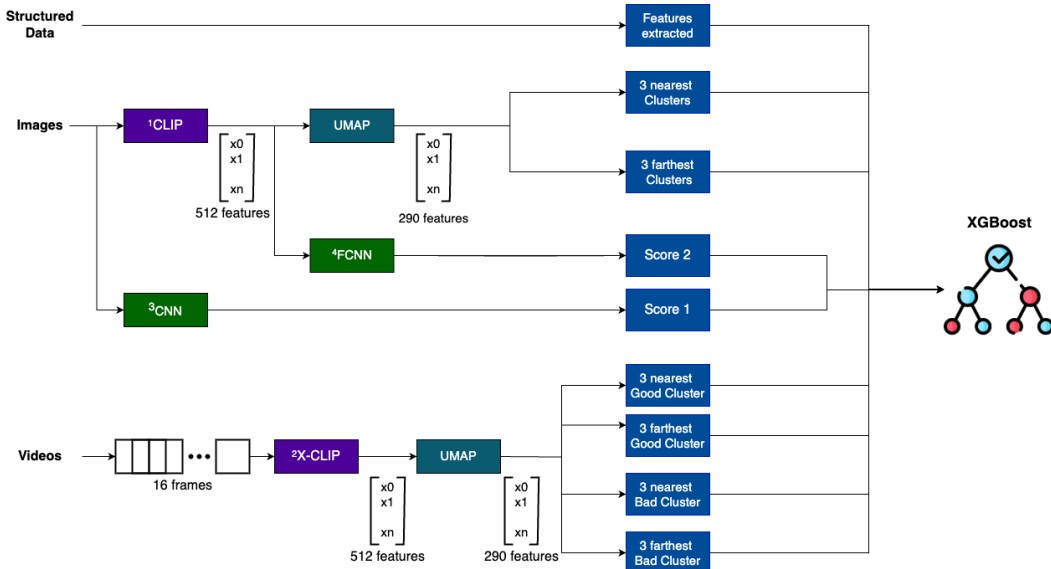

Figure 1: Pipeline of Instagram data extraction for XGBoost scoring system

criteria widely adopted in credit risk modeling. The results of these evaluations will clarify the effectiveness of incorporating computer vision-based features within credit risk assessment frameworks for informal micro-businesses.

## 2.2 IMAGE CLUSTERING USING CLIP EMBEDDINGS

Initially, images are resized to 224x224 pixels, centered, and normalized before feature extraction. Using CLIP model, a pretrained vision-language embedding generator, each image yields a 512-dimensional feature vector. The feature vectors are standardized employing the StandardScaler technique and subsequently subjected to dimensionality reduction through UMAP (refer to Table 1 for the parameter specifications of the UMAP model).

Table 1: UMAP Model Parameters for Dimensionality Reduction

| UMAP Parameter | Value |
|---|---|
| n_neighbors | 60 |
| min_dist | 0.25 |
| n_components | 290 |
| metric | euclidean |

Subsequently, the reduced features are clustered using the KMeans algorithm. To determine the optimal number of clusters, the Calinski-Harabasz Score, also known as the Variance Ratio Criterion (VRC), proposed by Calinski and Harabasz Caliński & Harabasz (1974), is computed for a range of cluster numbers. The VRC metric evaluates clustering quality by measuring the ratio of between-cluster dispersion to within-cluster dispersion, defined in Equation 2:

$$\text{VRC} = \frac{\frac{B(k)}{k-1}}{\frac{W(k)}{n-k}} \qquad (2)$$

where $B(k)$ denotes between-cluster dispersion, $W(k)$ represents within-cluster dispersion, $k$ is the number of clusters, and $n$ is the total number of data points.

For each image, distances to cluster centroids are calculated, and the three nearest and three farthest clusters are selected, yielding six categorical features integrated into the final predictive model.

## 2.3 Video Clustering Using X-CLIP Embeddings

The video analysis mirrors the image-based approach, using X-CLIP to extract 512-dimensional embeddings from 16 uniformly sampled frames per video. These are reduced and clustered as before. Clusters are labeled "good" or "bad" based on predominant client payment behavior in the training data. For each video, distances to all centroids are computed to identify the three closest and three farthest clusters in each category, yielding twelve categorical features.

This captures both similarity to reliable payers and dissimilarity from risky ones, leveraging the embedding space to improve prediction accuracy.

## 2.4 Neural Scoring Models

To extract predictive insights from visual content, we deploy two complementary neural network architectures. The first is a CNN that analyzes Instagram images directly, learning patterns indicative of payment behavior. Input images are resized to 224×224 pixels, center-cropped, and normalized using ImageNet statistics. The CNN architecture consists of standard convolutional layers, pooling operations, and fully connected layers, culminating in a binary classification output. The CNN outputs a probability score (Fig 1, Score 1) indicating repayment likelihood based solely on visual cues.

The second model is a FCNN that refines 512-dimensional CLIP embeddings into semantic-based risk scores. The arquitecture comprises an input linear layer (512 to 32 dimensions), three dropout layers (probabilities: 0.98, 0.95, and 0.90) to prevent overfitting, two sequential hidden layers (32 to 16, then 16 to 4 dimensions), and a final output linear layer (4 to 1 dimension). Activation functions include ELU with alpha=1.0 for non-linearity, and a softmax activation in the output layer to produce the probabilistic score (Fig 1, Score 2) capturing high-level visual semantics.

Both scoring models are implemented in PyTorch and deployed simultaneously through our API. This dual-scoring approach provides complementary perspectives on visual content: the CNN model captures direct visual patterns while the CLIP-based model leverages deeper semantic understanding.

## 3 Results

### 3.1 Evaluation of Image and Video Clustering

The Calinski-Harabasz Score was computed across a range of cluster numbers from 10 to 100 with increments of 5 to identify the optimal number of clusters. The optimal number of clusters identified was 40, corresponding to the highest observed VRC score (see Figure 2).

Following cluster determination, a qualitative evaluation of the clustering results was conducted through visual inspection. Examples of image clusters are presented in Figure 3. Cluster 16 predominantly grouped images related to food businesses, including fast food, traditional dishes, and diverse culinary presentations, while Cluster 0 grouped images largely related to cakes, pastries, and decorative arrangements typically associated with event planning and celebrations, demonstrating strong visual coherence within each cluster.

A similar clustering procedure was applied to the video embeddings extracted using X-CLIP, resulting in an optimal number of 80 clusters, also determined through evaluation of the Calinski-Harabasz Score and internal cluster consistency.

### 3.2 Impact of Computer Vision Features on the Final XGBoost Model

To evaluate the impact of incorporating computer vision-derived features, two XGBoost models were compared using performance metrics computed on the test set. The baseline model was trained exclusively on features extracted from Instagram user metadata and post interactions. The enhanced

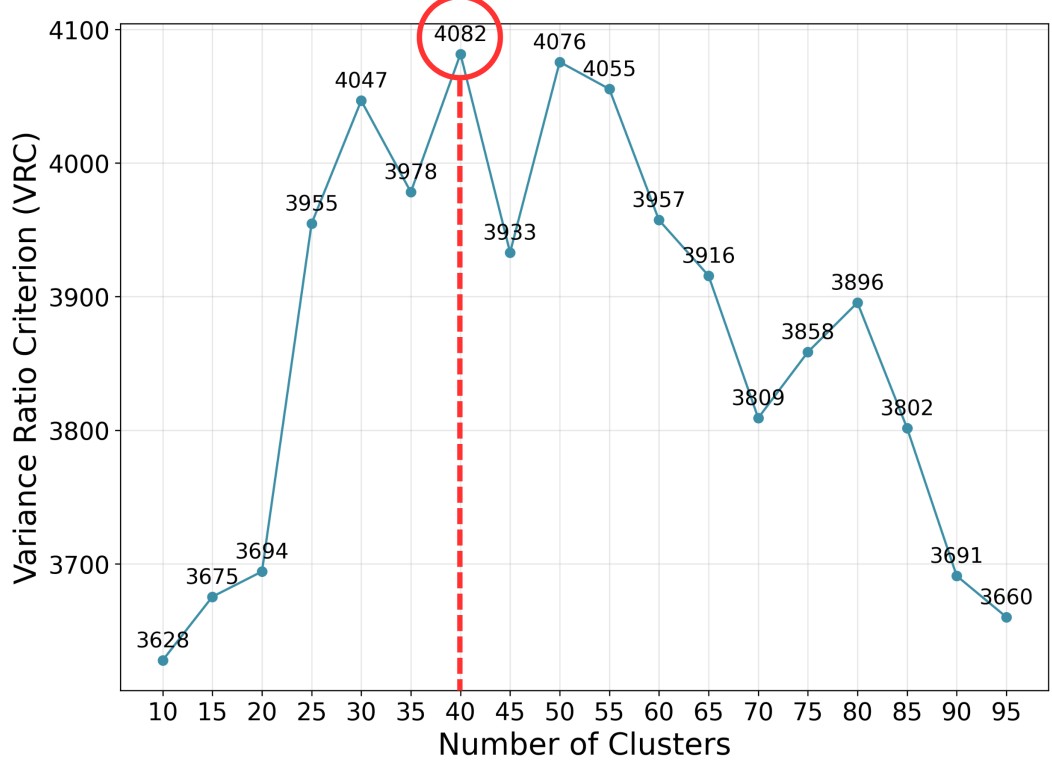

Figure 2: VRC scores for different numbers of clusters used in the unsupervised analysis of images.

model integrated these traditional features with additional visual features extracted from images and videos, as previously described.

The comparison of model performance in terms of AUC score and F1-score is summarized in Table 2.

Table 2: Performance Comparison of XGBoost Models with and without Visual Features

| Model Features | AUC (%) | F1-score (%) |
|---|---|---|
| Baseline features | 78.72 | 62.86 |
| Baseline + visual features | **80.88** | **72.72** |

As indicated in Table 2, incorporating visual features extracted from images and videos significantly improved the model performance, yielding an increase of 2.16 percentage points in AUC and a notable improvement of 9.86 percentage points in the F1-score.

Additionally, analysis of feature importance within the enhanced model revealed that visual features accounted for approximately 25.52% of the overall predictive power, underscoring their substantial contribution to the model's improved performance.

## 4    DISCUSSION AND FUTURE WORK

Our results indicate that image- and video-based embeddings add roughly 25% predictive lift on top of a baseline that already leverages caption text, hashtags, and posting-time data. This boost is promising but stems from a modest, Colombia-centric sample (570 firms; 44 in test) and a binary good/poor label, both of which curb statistical confidence and blur finer shades of credit behaviour.

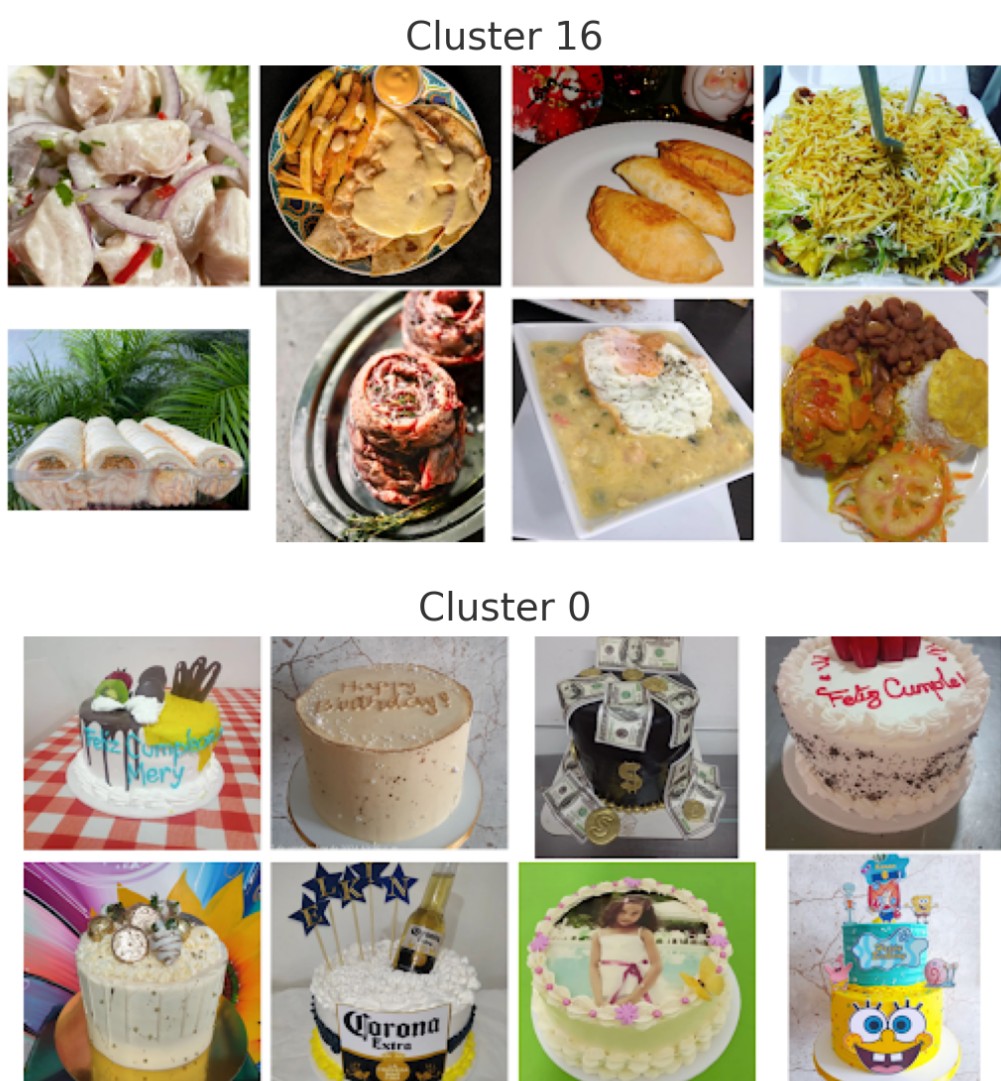

Figure 3: Sample images from two clusters formed by the KMeans algorithm. Each cluster contains business-related posts images from Instagram grouped by visual similarity.

Future work will broaden the dataset to other Latin-American markets and platforms, adopt multi-level delinquency labels (e.g., days-past-due bands), and run targeted ablations to isolate the incremental value of each visual modality and clustering setting—steps needed before moving toward production-grade scoring.

## 5 CONCLUSION

This study demonstrates that visual features from Instagram can significantly enhance credit scoring for informal microbusinesses. Visual embeddings improved AUC by 2.16 and F1-score by 9.86 percentage points over metadata-only models. Feature importance analysis revealed that visual features contributed 25.52% of the model's predictive power, highlighting their value in capturing business traits relevant to creditworthiness. Thus, computer vision techniques applied to social media content offer a promising pathway for fintech companies aiming to improve financial inclusion by more accurately predicting credit risk among underserved entrepreneurial communities.

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
