# OpenReview forum: "Leveraging Visual Embeddings from Instagram for Credit Scoring of Informal Microbusinesses"
_ICLR.cc/2026/Conference — Submitted to ICLR 2026_

### Official Review · Reviewer_o2Fb · 2025-10-30

**Soundness:** 2
**Presentation:** 3
**Contribution:** 2
**Rating:** 4
**Confidence:** 3

**Summary:**

This paper addresses the challenge of limited access to formal credit for informal microbusinesses in Latin America. Many entrepreneurs in this sector resort to predatory lending practices characterized by exorbitant interest rates. To address this, the authors propose a novel approach that leverages computer vision techniques applied to Instagram images and videos from microbusiness accounts to improve creditworthiness prediction models.

**Strengths:**

1. Novelty and Significance : The paper proposes a novel approach to credit scoring by integrating computer vision pipelines applied to Instagram data. This is particularly important because it targets informal microbusinesses in Latin America, addressing a substantial credit gap. Leveraging digital footprints and visual semantics to assess credit risk is an innovative application of representation learning.
2. Empirical Results : The study provides clear evidence of significant performance enhancement, notably an increase of 2.16 percentage points in AUC and a substantial 9.86 percentage points in F1-score. Furthermore, the finding that visual features account for 25.52% of the overall predictive power underscores their substantial contribution to enhanced model performance.
3. Robust and Complex Technical Methodology : The methodology is technically sophisticated, combining multiple machine learning techniques: using pre-trained vision-language models, UMAP and KMeans to derive features, and deploying CNN and FCNN to generate dual visual scores. The use of a custom evaluation metric that prioritizes AUC maximization while penalizing high default rates and encouraging class separation  demonstrates a deep understanding of practical credit risk objectives.

**Weaknesses:**

1. Limited Sample Size and Generalization Constraints : The results stem from a modest, Colombia-centric sample of only 570 microbusinesses. The test set is particularly small, consisting of only 44 users, which the authors acknowledge that this ‘limits statistical reliability’. This severely limits the statistical reliability and generalizability of the findings across Latin America or other platforms.
2. Coarse Granularity of Credit Behavior Labels : The dataset uses a simple binary classification system, labeling borrowers as either 'good payer' or 'poor payer'. The authors concede that this binary approach may blur finer shades of credit behavior. They recommend, as future work, the adoption of multi-level delinquency labels to capture credit behavior more accurately.
3. Potential Sector-Selection Bias : The dataset exhibits an uneven distribution of business categories. The largest categories each account for 19%, while Agriculture and Electronics combined account for less than 5%. This high concentration in specific sectors may introduce a sector-selection bias, making the model's performance potentially less reliable for underrepresented industries.

**Questions:**

1. Custom Evaluation Metric Weights and Sensitivity : The custom evaluation metric used for optimizing the XGBoost model includes four weights. What were the specific numerical values chosen for these weights during the hyperparameter tuning process? Furthermore, were any sensitivity analyses performed to assess how changes in these weights impacted the final performance metrics?
2. Contribution of Visual Modalities : Visual features contributed 25.52% of the total predictive power. The proposed methodology utilizes both clustering-derived features and neural scores. Can the authors provide a targeted breakdown: Which of these visual components contributed the most to the overall 25.52% improvement?
3. Justification for Extreme FCNN Dropout Rates : The FCNN architecture implemented to refine CLIP embeddings uses exceptionally high dropout probabilities in its layers to prevent overfitting. Given the small size of the training set and the validation set, this level of regularization seems extreme. What experimental evidence or specific metrics justified the adoption of such high dropout rates, and how was the potential loss of model capacity assessed?
4. Embedding Space Alignment : The approach uses CLIP for images and X-CLIP for videos, both yielding 512-dimensional embeddings. Were these two distinct embedding spaces subject to any alignment or normalization process before the features were concatenated and integrated into the final XGBoost model? Additionally, why was the optimal number of clusters significantly different (40 for images versus 80 for videos)?
5. Temporal Performance and Concept Drift: The dataset uses an out-of-time strategy across a 2022–2024 period to ensure temporal validity. Considering economic and social changes over two years, there is a risk of concept drift. What analysis was conducted to confirm that the model, trained on older data, maintained stable or consistent predictive performance when tested specifically on the newest subset of data in the test set?

---

### Official Review · Reviewer_gq6a · 2025-10-31

**Soundness:** 2
**Presentation:** 2
**Contribution:** 2
**Rating:** 2
**Confidence:** 3

**Summary:**

This paper proposes using visual embeddings from Instagram posts to enhance credit scoring for informal microbusinesses in Latin America. The method extract features using CLIP models, apply dimensionality reduction and clustering, and feed these into XGBoost for creditworthiness prediction.

**Strengths:**

+ financial inclusion for underserved communities is a significant societal challenge worth addressing.

**Weaknesses:**

- Very limited novelty, it simply applies off the shelf CLIP models without any domain adaptation, with standard UMAP and KMeans clustering pipeline. There is no clear justification for architectural choices.
- Limited ablation studies to validate design decisions.
- The dataset is very limited, only 570 samples with 44 samples in the test set. The claimed improvements fall well within statistical noise given this sample size, with no confidence intervals, significance tests, or cross-validation reported.
- There are severe presentation issues, like writing quality, and limited quality figures.

**Questions:**

- What measures have you taken to prevent discriminatory lending based on visual aesthetics?

**Details Of Ethics Concerns:**

- Using visual content to determine creditworthiness may lead to discrimination risks. Could perpetuate biases against businesses with less polished visual presentations.
- There is limited discussion of fairness, demographic parity, or potential for discrimination.

---

### Official Review · Reviewer_EU26 · 2025-11-02

**Soundness:** 2
**Presentation:** 3
**Contribution:** 2
**Rating:** 2
**Confidence:** 3

**Summary:**

This paper presents a methodology for credit scoring of informal microbusinesses in Latin America using visual data from their Instagram profiles. The authors extract features from images and videos via pre-trained models (CLIP, X-CLIP) and two supervised neural networks (a CNN and an FCNN). These visually-derived features are then combined with account metadata in an XGBoost model to predict loan repayment behavior. On a small dataset from a Colombian fintech company, the authors report that adding visual features improves the AUC by 2.16 points and the F1-score by 9.86 points compared to a baseline using only metadata.

**Strengths:**

The paper tackles a impactful real-world challenge—improving financial inclusion for underserved entrepreneurs. The core idea of using visual information from social media as a proxy for business health is creative and explores a promising direction for credit scoring in data-scarce environments.

**Weaknesses:**

1.A test set of only 44 users is insufficient to draw robust conclusions. The reported performance metrics could be subject to high variance and may not hold on a larger, more diverse sample.
2.The methodology combines pre-trained embeddings, dimensionality reduction, clustering, and two separate deep learning models into a final XGBoost model. The contribution of each part is unclear.
3. The model may be learning spurious correlations instead of true signals of creditworthiness. For example, it might associate high-quality photos (requiring an expensive phone or camera) with good payers, thus creating a model that discriminates based on the socio-economic status of the business owner rather than their business's viability. This critical aspect is not adequately addressed.

**Questions:**

1. Why were the 3 nearest and 3 farthest clusters chosen as features? How sensitive is the model to this number?
2.The custom evaluation metric (Equation 1) introduces complexity without a clear comparison to optimizing for a standard metric like AUC or F1-score directly.
3. Is it correct to assume that at inference time for a new client, you would not know if their videos belong to a "good" or "bad" cluster a priori? How is this handled for new data?

---

### Meta-Review · Area_Chair_s1ib · 2026-01-07

**Summary:**

This paper was reviewed by three reviewers and received three negative recommendations. The major concerns are
- limited novelty
- insufficient experiments and ablation studies
- unclear contribution.

**Reviewer Concerns:**

The authors did not provide responses to the comments.

**Reviewer Scores:**

The reviewers would not have changed their scores since there was no author discussion and the paper should be rejected.

---

### Decision · Program_Chairs · 2026-01-26

Reject